# Benzo[*f*]indole-4,9-dione Derivatives Effectively Inhibit the Growth of Triple-Negative Breast Cancer

**DOI:** 10.3390/molecules26154414

**Published:** 2021-07-21

**Authors:** Fabiana Sélos Guerra, Flaviana Rodrigues Fintelman Dias, Anna Claudia Cunha, Patricia Dias Fernandes

**Affiliations:** 1Laboratório de Farmacologia da Dor e da Inflamação, Instituto de Ciências Biomédicas, Universidade Federal do Rio de Janeiro, Rio de Janeiro 21941-901, Brazil; patricia.dias@icb.ufrj.br; 2Programa de Pós-graduação em Farmacologia e Química Medicinal, Instituto de Ciências Biomédicas, Universidade Federal do Rio de Janeiro, Rio de Janeiro 21949-900, Brazil; 3Departamento de Química Orgânica, Instituto de Química, Campus do Valonguinho, Universidade Federal Fluminense, Niterói 24020-140, Brazil; flavianarfd@id.uff.br (F.R.F.D.); annaclacbio@gmail.com (A.C.C.)

**Keywords:** benzo[*f*]indole-4,9-dione, cancer, breast cancer, triple-negative breast cancer, apoptosis, cell cycle

## Abstract

Triple-negative breast cancer (TNBC) is a subtype of breast cancer with poor clinical outcome, and currently no effective targeted therapies are available. Indole compounds have been shown to have potential antitumor activity against various cancer cells. In the present study, we found that new four benzo[*f*]indole-4,9-dione derivatives reduce TNBC cell viability by reactive oxygen species (ROS) accumulation stress in vitro. Further analyses showed that LACBio1, LACBio2, LACBio3 and LACBio4 exert cytotoxic effects on MDA-MB 231 cancer cell line by inducing the intrinsic apoptosis pathway, activating caspase 9 and Bax/Bcl-2 pathway in vitro. These results provide evidence that these new four benzo[*f*]indole-4,9-dione derivatives could be potential therapeutic agents against TNBC by promoting ROS stress-mediated apoptosis through intrinsic-pathway caspase activation.

## 1. Introduction

Breast cancer is a heterogeneous disease with varied biological features and histology, and it is the second most common cause of cancer mortality in women worldwide [1,2]. Triple-negative breast cancer (TNBC) is an aggressive subtype that frequently develops resistance to chemotherapy. In this subtype, estrogen and progesterone receptors are not expressed and human epidermal growth factor receptor 2 is not overexpressed [3,4]. TNBC is correlated with poor prognosis, metastases, recurrence and high mortality rates despite systemic therapy [5]. Treating patients with TNBC remains clinically challenging.

Quinones represent a class of natural and synthetic compounds that have many beneficial effects, including antifungal, antiprotozoal, antibacterial and anticancer activities [6,7]. Some naturally occurring quinones also have anticancer properties and are widely used as frontline chemotherapeutic agents. Indolequinones are also potent inhibitors of the growth of human pancreatic cancer cells, suggesting a potential role for such compounds as therapeutic agents [8]. The cytotoxicity of quinone compounds has been described by two general mechanisms. One of these mechanisms mediated via effects on biomolecules (DNA, RNA, lipids and proteins), and the other is mediated through the production of reactive oxygen species (ROS), particularly hydrogen peroxide and hydroxyl radicals [9,10].

ROS are known physiological effectors of cell signaling and mitogenic pathways and increase cytotoxic activity in cancer cells by inducing apoptosis [11,12]. ROS can interact with lipids, proteins, RNA and DNA and cause irreversible damage to these molecules [13]. However, the oxidative stress induced by indole derivatives is not well understood.

We have previously synthesized three series of carbohydrate-based benzo[*f*]índole-4,9-diones and amino-1,4-naphthoquinone derivatives and evaluated their cytotoxic activity by MTT assay against eight human cancer cell lines and nontumor human erythrocytes [14]. From this study, we choose four benzo[*f*]indole-4,9-dione derivatives, LACBio1, LACBio2, LACBio3 and LACBio4, and evaluated their action against the TNBC cells MDA-MB 231. We found that the four compounds studied were able to induce apoptosis in MDA-MB 231 cells through the activation of proapoptotic proteins and induce G2/M cell cycle arrest. We also found that the compounds LACBio1, LACBio2 and LACBio3 increased the total ROS level, and this increase led to the activation of proapoptotic pathways. Furthermore, our studies suggest that these four benzo[*f*]indole-4,9-dione derivatives are potential prototypes for the development of drugs for the treatment of TNBC.

## 2. Results

### 2.1. Benzo[f]indole-4,9-dione Derivatives Induced Apoptosis in MDA-MB 231 Cells

To understand the mechanisms involved in cell death, we carried out annexin V–PI double staining. The annexin V-FITC/PI binding assay detects live cells (Q2-LL; AV−/PI−), early apoptotic cells (Q2-LR; AV+/PI−), late apoptotic cells (Q2-UR; AV+/PI+) and necrotic cells (Q2-UL; AV−/PI+). MDA-MB 231 cells were treated with 10 and 30 µM of LACBio1, LACBio2, LACBio3 and LACBio4 for 24 h or untreated (control) and stained with annexin V-FITC and propidium iodide. At all concentrations tested, LACBio1, LACBio2, LACBio3 and LACBio4 caused 22–25% early apoptosis and up to 17–21% late apoptosis of MDA-MB 231 cells within 24 h of treatment (Figure 1a,b). In contrast, nontreated cells (control) showed about 90% live cells, 8.6% early apoptosis and 0.93% late apoptosis. These results suggest that LACBio1, LACBio2, LACBio3 and LACBio4 induced apoptosis in MDA-MB 231 cells.

### 2.2. Benzo[f]indole-4,9-dione Derivatives Caused DNA Fragmentation in MDA-MB 231 Cells

DNA fragmentation represents a characteristic hallmark of apoptosis, and the TUNEL assay is a well-known method for detecting such DNA fragments. Apoptosis can be initiated by extrinsic and intrinsic pathways, which both lead to a final common pathway and DNA fragmentation [15]. To understand the mechanisms involved in cell apoptosis, we carried out ApopTag TUNEL assay–PI double staining. Figure 2 shows that nontreated cells (control) presented normal DNA without significant DNA damage. In contrast, all the concentrations tested for quinone derivatives induced DNA fragmentation (at 80–90% levels) within 24 h of treatment.

### 2.3. Effects of Benzo[f]indole-4,9-dione Derivatives on Cell Cycle Distribution

Next, we studied the effects of LACBio1, LACBio2, LACBio3 and LACBio4 in their capacity to induce alterations in cell cycle progression. As shown in Table 1, preincubation of MDA-MB 231 cells with the compounds induced cell cycle arrest at the G2/M phase.

### 2.4. Benzo[f]indole-4,9-dione Derivatives Increased Intracellular Reactive Oxygen Species (ROS) Levels

To investigate whether ROS were involved in benzo[*f*]indole-4,9-dione derivative-mediated cell death, we next measured ROS levels within the cells by using a ROS-sensitive fluorometric probe, DCFH-DA. As shown in Figure 3, LACBio1, LACBio2 and LACBio3 (at 10 and 30 µM) induced a significant increase in the generation of ROS after 1 h of exposure while LACBio4 caused significant increases in the generation of ROS only at 10 µM. It is interesting to note that compounds induced an increase in ROS production at levels similar to the positive control group (H_2_O_2_).

### 2.5. Benzo[f]indole-4,9-dione Derivatives Increased the Bax:Bcl-2 Ratio

Alterations in the expression of Bcl-2 family proteins regulate the commitment of cells to apoptosis. We performed Western blotting to analyze the expression of Bax (proapoptotic) and Bcl-2 (antiapoptotic) proteins and to determine an increase in the Bax:Bcl-2 ratio. Figure 4 shows that cells treated with LACBio1, LACBio2 and LACBio4 showed an increase in Bax and a decrease in Bcl-2 expression (Figure 4a). The Bax:Bcl-2 ratio was significantly increased by almost 75% after treatment with 30 µM of LACBio1, LACBio2 and LACBio4 compared with the control group. The increase in the Bax:Bcl-2 ratio due to benzo[*f*]indole-4,9-dione derivative treatments indicated a commitment of the cells to apoptosis via the mitochondrial release of proapoptotic molecules (Figure 4).

### 2.6. Benzo[f]indole-4,9-dione Derivatives Induced Activation of the Intrinsic Caspase Cascade

In the next step, we performed a Western blotting analysis to examine the possible activation of caspases 3 and 9 in MDA-MB 231 cells following LACBio1, LACBio2, LACBio3 and LACBio4 treatment. Figure 5a shows representative data obtained after incubation of cells with LACBio1, LACBio2, LACBio3 and LACBio4 in the expression of caspases 3 and 9 and their cleaved forms. When the data were expressed as arbitrary units obtained from different experiments, it could be noted that LACBio1, LACBio2 and LACBio3 (at 10 or 30 µM) caused a significant increase in caspase 3 and 9 activity, which was observed as an increase in the levels of cleaved caspase (Figure 5b,c, respectively). LACBio4 caused significant increases in caspase 9 and caspase 3 only at a concentration of 10 µM. These results suggest that the compound-mediated caspase 9 activation could in turn activate effector caspase 3 for apoptosis in MDA-MB 231 cells.

## 3. Discussion

The clinical prognosis of triple-negative breast cancer (TNBC) is worse than that of other types of breast cancer. It is a more aggressive high-grade breast cancer and has a characteristic absence of sensitivity to current targeted drugs [16].

In the present study, we evaluated four benzo[*f*]indole-4,9-dione derivatives against TNBC cells. Our data demonstrate that LACBio1, LACBio2, LACBio3 and LACBio4 induce apoptosis in MDA-MB 231 cells via activation of intrinsic caspase pathways by increasing intracellular ROS levels.

Our current data demonstrate that all compounds significantly inhibited proliferation and induced apoptosis in MDA-MB 231 cells. Consistent with our prior study [14], we showed that benzo[*f*]indole-4,9-dione derivatives are promising therapeutic agents for the induction of apoptosis, specifically in highly proliferative and metabolically active TNBC cells, without seriously affecting normal cells. We could see that the four structurally related indoles showed similar biological patterns. The results demonstrate that indole derivatives (at 10 and 30 μM) decreased cell viability and increased the number of cells undergoing cell death from apoptosis (early or late apoptosis) compared to the control.

Although benzo[*f*]indole-4,9-dione derivatives have shown the capacity to activate the apoptotic pathway, we examined their effects on DNA fragmentation. DNA fragmentation, the end phase of apoptosis, is the ultimate step in cell death [17]. Our data demonstrate that irrespective of the substitution pattern of the indole ring, the compounds were capable of inducing apoptotic DNA fragmentation in human MDA-MB 231 cells. In addition, these results corroborate those from Gach and colleagues [18], who showed the cytotoxic activity of naphthofuran-4,9-diones and benzoindole-4,9-diones by demonstrating their ability to inhibit cell proliferation and DNA damage and induce apoptosis in the leukemia cell line HL-60 and the breast adenocarcinoma MCF-7 cell line.

The blockade of cell cycle progression by chemotherapeutic agents has always been an ideal choice for developing anticancer therapeutics, and many cytotoxic compounds exert their inhibitory effect by arresting the cell cycle at a specific checkpoint [19]. Noroozi and colleagues [20] analyzed the effects of two indole compounds on the inhibition of proliferating cells in acute promyelocytic leukemia (NB4 cell line) by examining the cell cycle, and they demonstrated that both indoles induced apoptosis, thereby facilitating cell cycle arrest. In our work, we found that the four benzo[*f*]indole-4,9-dione derivatives induced cell cycle arrest at the G2/M phase. The first checkpoint in the cell cycle is observed at the G1/S boundary, and the second occurs at the G2/M transition. These checkpoints control the mechanisms that ensure the proper timing of cell cycle events [21]. Entry to the mitosis phase is blocked by the G2 checkpoint mechanism when DNA is damaged [22]. Our results provide evidence that all benzo[*f*]indole-4,9-dione derivatives, regardless of their concentration, can cause DNA damage, thereby leading to cell cycle arrest and subsequent cell death through apoptosis.

Reactive oxygen species (ROS) are well recognized as mediators of DNA damage [23]. Chemotherapeutics, such as doxorubicin and cisplatin, increase ROS levels, thus contributing to their genotoxicity [24,25]. Our data showed that all quinone benzo[*f*]indole-4,9-dione derivatives increased ROS production by MDA-MB 231 cells, suggesting that it could be the cause of DNA fragmentation and consequently cell death. Indole compounds LACBio1, LACBio2 and LACBio3 were more cytotoxic at the two concentrations under study, and they increased ROS production in the dichlorodihydro fluorescein diacetate assay and induced apoptosis in MDA-MB 231 cancer cells, while the derivative LACBio4 increased reactive oxygen species levels in the cells upon treatment with 10 µM. The compounds LACBio1 and LACBio2, which have an amino group at the C-2 position of the quinone nucleus, were more effective at generating reactive oxygen species in MDA-MB 231 cells than the other derivatives tested. This result is in agreement with the data reported in the literature, which indicate that the introduction of oxidizable groups, such as amino substituent, into the quinone ring, can exert an influence on its redox properties [26]. We suggest that the solubility of LACBio4 at the highest concentration, 30 µM, in the ROS generation assay could interfere with the results, requiring a treatment time longer than 1 h, as established in the assay, for this compound to reach its site of action in the cell.

ROS have also been suggested to regulate the process involved in the initiation of apoptotic signaling [27]. Quast and colleagues [28] demonstrated that abrogated ROS production suppressed apoptosis and Bax activation. Bax is a member of the Bcl-2 family and a regulator protein in mitochondrial proapoptotic pathways. Proteins of the Bcl-2 family have been shown to play an important role in the regulation of mitochondrial-mediated apoptosis [29]. It has also been reported that Bcl-2 suppresses ROS-induced apoptosis [30] and that the overexpression of proapoptotic Bax enhances ROS generation [31]. The results of our study showed that the treatment of cells with indole derivatives LACBio1, LACBio2 and LACBio4 altered the balance between proapoptotic Bax and antiapoptotic Bcl-2 proteins at the mitochondrial membrane. This effect was not observed for the related indole derivative LACBio3. For the compounds LACBio1, LACBio2 and LACBio4, these results suggest that the increased expression of the two antiapoptotic proteins can be related to the chemical structures (e.g., conformation and intermolecular interactions) of the pyranose and furanose rings.

Apoptosis is a genetically regulated biological process with two major pathways: the extrinsic pathway, induced by the activation of the death receptor, and the intrinsic pathway, mediated by mitochondrial apoptosomes [32]. The intrinsic mitochondrial pathway is activated by a variety of exogenous and endogenous stimuli, including oxidative stress, DNA damage and ischemia [33]. This pathway is influenced by members of the Bcl-2 family bound to the mitochondrial membrane, including Bax and Bcl-2 [34]. Additionally, intrinsic pathway activation ultimately triggers mitochondrial outer membrane permeabilization and can facilitate the release of several proapoptotic factors, including cytochrome c, from the mitochondrial intermembrane space into the cytoplasm. These proapoptotic factors can activate caspase 9, activate the effector caspase 3 and induce cell death [35]. The compounds LACBio1, LACBio2, and LACBio3 were capable of activating caspase 9 and caspase 3 at the two concentrations tested, while for the derivative LACBio4, only the lowest concentration (10 µM) showed this effect. These results confirm that indole derivatives caused an increase in intracellular ROS, resulting in damage to DNA and mitochondria and consequently inducing cells to undergo apoptosis, probably via the intrinsic route.

## 4. Materials and Methods

### 4.1. Synthesis of Benzo[f]indole-4,9-dione Derivatives

The compounds LACBio1, LACBio2, LACBio3 and LACBio4 investigated in this work were synthesized as previously described by Dias and colleagues [14]. The synthesis of the compounds LACBio1 and LACBio2 involved the reaction of halo-naphthoquinones containing a carbonyl group with amino carbohydrates. The compounds LACBio3 and LACBio4 (Scheme 1) were prepared by a cerium(IV)-mediated oxidative free radical cyclization reaction between 1,4-amino-naphthoquinones, which possess an aminocarbohydrate chain at the C-2 position of the quinone ring, and ethyl acetoacetate [36].

### 4.2. Cell Culture and Treatment

The human breast cancer cell line MDA-MB 231 was obtained from the American Type Culture Collection (ATCC HTB-26). Cells were routinely grown in Roswell Park Memorial Institute (RPMI) essential medium (Sigma Aldrich, St. Louis, MO, USA) containing 10% fetal bovine serum (LGC Biotechnology, Cotia, SP, Brazil) and 1% penicillin–streptomycin (Sigma Aldrich, St. Louis, MO, USA) in a humidified 5% CO_2_ atmosphere at 37 °C. Cells were cultured up to 70–100% confluence. The compounds were dissolved in dimethyl sulfoxide (DMSO) (Sigma Aldrich, St. Louis, MO, USA) in a 1 mM stock solution, and the concentrations used were dissolved in RPMI at different concentrations. The DMSO concentration did not exceed 0.1% in the final solution and had no effect per se.

### 4.3. Apoptosis Assay

Apoptosis was carried out by an annexin V staining kit (BD Biosciences, San Jose, CA, USA) according to the manufacturer’s protocol. Briefly, cells (2.5 × 10^5^ cells/mL) were seeded into 12-well plates and incubated with the compounds (at 10 or 30 μM) for 24 h. After the incubation period, the cells were harvested and washed with phosphate-buffered saline (PBS) and resuspended in the indicated binding buffer. Subsequently, annexin V-FITC and propidium iodide (PI) were added to the cell suspension and incubated for 15 min in the dark at room temperature. After incubation, 10^4^ events/sample were analyzed by flow cytometry (BD Accuri C6 Plus, BD Biosciences, San Jose, CA, USA), and the data were analyzed by BD CFlow Plus software (BD Biosciences, San Jose, CA, USA).

### 4.4. DNA Fragmentation Analyses

To detect DNA fragmentation in MDA-MB 231 cells, a modified TUNEL method [37] was used with the ApopTag Fluorescein In Situ Apoptosis Detection Kit (Merck, KGaA, Darmstadt, Germany). Briefly, MDA-MB 231 cells were grown in 12-well culture plates and incubated with the compounds (10 or 30 μM) for 24 h. Following treatments, the cells were washed with PBS and subsequently stained with ApopTag according to the manufacturer’s instructions. Experiments were conducted in triplicate, 10^4^ events/sample were analyzed by flow cytometry (BD Accuri C6 Plus, BD Biosciences, San Jose, CA, USA), and data were analyzed by BD CFlow Plus software (BD Biosciences, San Jose, CA, USA).

### 4.5. Cell Cycle Analysis

The cell cycle was assessed using a BD Biosciences flow cytometry kit according to the manufacturer’s protocol (BD Biosciences, San Jose, CA, USA). Cells were seeded at 2.5 × 10^5^ cells/well in 12-well plates and incubated with the compounds (10 or 30 μM) at 37 °C for 24 h in a humidified chamber containing 5% CO_2_. After staining the samples with PI (500 µg/mL) for 15 min, 10^4^ events were analyzed by flow cytometry (BD Biosciences, San Jose, CA, USA), and the data were analyzed by BD CFlow Plus software (BD Biosciences, San Jose, CA, USA).

### 4.6. ROS Generation Assay

The intracellular generation of ROS was assayed with the 2′,7′-dichlorofluorescein diacetate (DCFH-DA) method described by Eruslanov and Kusmartsev [38]. MDA-MB 231 cells were seeded at 5 × 10^4^ cells/well in a 12-well plate and precultured overnight. The next day, the cells were preincubated with DCFH-DA (5 μM) in RPMI for 30 min. Then, the cells were incubated with compounds (10 or 30 μM) or with H_2_O_2_ (0.03%, as the positive control) for 1 h. After incubation, the cells were washed with PBS, and the level of DCFH-DA fluorescence was determined using a Flexstation 3 microplate reader at 485 nm absorption and 535 nm transmission (Molecular Devices, San Jose, CA, USA).

### 4.7. Western Blotting Analysis

The protein content of the cell lysate was determined by a bicinchoninic acid (BCA) protein assay reagent kit (Thermo Fisher Scientific, MA, USA) according to the manufacturer’s protocol. Before the Western blot analysis, protein samples (20 μg) were subjected to SDS-PAGE and stained with Laemmli buffer to ensure equal loading of the proteins. Proteins were transferred onto a nitrocellulose membrane (Amersham Hybond-C extra; GE Healthcare, Chicago, IL, USA). The membranes were incubated overnight at 4 °C with primary antibodies and subsequently with secondary rabbit horseradish peroxidase antibodies for 1 h at room temperature. The following primary and secondary antibody combinations were used for the proteins: caspase 3, caspase 9, Bax, Bcl-2 and HRP-conjugated anti-rabbit IgG were obtained from Cell Signaling Technology (Beverly, MA, USA) and β-actin was purchased from Merck/Sigma-Aldrich (Merck KGaA, Darmstadt, Germany).

### 4.8. Statistical Analysis

All the values are presented as the mean ± standard error of three independent experiments. Statistical analyses were performed via a one-way ANOVA with Dunnett´s post hoc test, and statistical significance was defined as * *p* < 0.05. Statistical analyses were performed with GraphPad Prism 8.02 (GraphPad Software Inc., San Diego, CA USA).

## 5. Conclusions

In conclusion, in the present study, we showed that LACBio1, LACBio2, LACBio3 and LACBio4 can induce cell death by apoptosis through an increase in ROS, thus causing DNA damage and inducing G2/M cell cycle arrest. Additionally, we show that the four benzo[*f*]indole-4,9-dione derivatives can induce cleavage of caspases (3 and 9), thereby activating the intrinsic pathway of apoptosis (Scheme 2).

Taken together, we suggest several new compounds as potent cytotoxic agents, and they provide new possibilities for developing a new series of compounds to be used in the treatment of TNBC and other cancer types.

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
