# Peer review of "Benzo[f]indole-4,9-dione Derivatives Effectively Inhibit the Growth of Triple-Negative Breast Cancer"

_molecules, 2021, doi:10.3390/molecules26154414_

Round 1

Reviewer 1 Report

In this study, the authors showed that four indole compounds have the cell cytotoxicity toward MDA-MB 231 cancer cell lines, these compounds can cause the increase ROS level of cells. Moreover, the authors showed that the cell apoptosis was initiated through the caspase 9 and BAX-Bcl-2 pathway. In general, there is no clear significance for the research . It is quite common to find compounds that can lead cell death via cell apoptosis. The molecular mechanism is simple and clear for cell apoptosis, there is nothing surprise to the audience. ALthough the authors claimed that treatment of TNBC, thesea data are far away to demonstrate this topics. The writing of the paper is clear, only the significance and the novel of this research was away from the current published level. Thus, I do not recommend the publiscation of this paper in Molecules. 

Author Response

Thank you very much for your review. The research and development of new drugs are deeply rooted linked to scientific and technological innovations. The search for formulations, biochemical pathways and molecular targets with remarkable therapeutics has been driving research and development work. We designed this study in order to provide additional information about the action and the mechanism of action underlying the cytotoxicity of 4 new compounds, benzo[f]indole-4,9-dione derivatives, against the TNBC cells MDA-MB 231, a kind of cancer that has fewer treatment options, and we have done it for the first time. We agree that the total explanation is not possible at this time and our conclusion is only suggested by our results. Inhibition of multiple targets of growth and survival pathways, especially the cell cycle and apoptosis, is an attractive strategy that can be used to eliminate cancer cells. The experimental design of this study follows the standards of the newest assays related to cytotoxicity and evaluation of apoptotic processes recently published in the scientific community (Sharma et al 2016; Jantamat et al., 2019; Atmaca et al., 2020). Our results showed that all the synthesized compounds exhibited a significant cytotoxic effect against TNBC cells MDA-MB 231 and, in conjunction with our previous studies, we showed that these compounds have no activity on normal human erythrocytes (Dias et al., 2020). Several statements that we made were more ambiguous than intended, and we have adjusted the text to be clearer.

Sharma P, Thummuri D, Reddy TS, Senwar KR, Naidu VGM, Srinivasulu G, Bharghava SK, Shankaraiah N. New (E)-1-alkyl-1H-benzo[d]imidazol-2-yl)methylene)indolin-2-ones: Synthesis, in vitro cytotoxicity evaluation and apoptosis inducing studies. Eur J Med Chem. 2016 Oct 21;122:584-600. doi: 10.1016/j.ejmech.2016.07.019. Epub 2016 Jul 11. PMID: 27448916.

Jantamat P, Weerapreeyakul N, Puthongking P. Cytotoxicity and Apoptosis Induction of Coumarins and Carbazole Alkaloids from Clausena harmandiana. Molecules. 2019 Sep 18;24(18):3385. doi: 10.3390/molecules24183385. PMID: 31540345; PMCID: PMC6767265.

Atmaca H, İlhan S, Batır MB, Pulat ÇÇ, Güner A, Bektaş H. Novel benzimidazole derivatives: Synthesis, in vitro cytotoxicity, apoptosis and cell cycle studies. Chem Biol Interact. 2020 Aug 25;327:109163. doi: 10.1016/j.cbi.2020.109163. Epub 2020 Jun 12. PMID: 32534988.

Dias, F.R.F., Guerra, F.S., Lima, F.A., Castro, Y.K.C., Ferreira, V.F., Campos, V.R., Fernandes, P.D., Cunha, A.C., 2020. Synthesis and biological evaluation of Benzo[f]indole-4,9-diones N-linked to carbohydrate chains as new type of antitumor agents. J. Braz. Chem. Soc. 2020, in press. 

Reviewer 2 Report

Triple-negative breast cancer (TNBC) is an aggressive breast cancer subtype that frequently develops resistance to chemotherapy, metastasizes early despite adjuvant treatment and carries a very poor prognosis. As TNBC has fewer treatment options than other types of invasive breast cancer, there is a critical need to discover new therapeutic agents to treat patients with TNBC.

In this study, Guerra et al. evaluate the action of four benzo[f]indole-4,9-dione derivatives against the TNBC cells MDA-MB 231. Authors show that the four benzo[f]indole-4,9-dione derivative compounds (LACBio1 to LACBio4) can induce cell death by apoptosis through an increase in the generation of ROS, thus causing DNA damage and inducing G2/M cell cycle arrest. Additionally, authors demonstrate that these compounds exert cytotoxic effects on TNBC by inducing intrinsic apoptosis pathway, activating BAX-Bcl-2 pathway and caspase 9 in vitro.

Globally, I find this study well designed, well performed and correctly interpreted. However, the following points should be addressed to merit publication in Molecules.

Major comments

  • In this study, the authors treated MDA-MC 231cells with the 4 compounds (at 10 or 30µM) for 24h. Please, could the authors explain why they selected such concentrations to treat the cells. Have the authors performed a viability test (such as a MTT assay) to determine the IC50 of the 4 compounds on these cells?

  • The authors demonstrate that the 4 indole derivatives induce apoptosis in MDA-MD 231 cells through an increase in ROS thus causing DNA fragmentation. Have the authors considered and tested other mechanisms involved in cell death, other than apoptosis, such as autophagy, necrosis or ferroptosis?

  • The authors show that four benzo[f]indole-4,9-dione derivatives inhibit the growth of triple-negative breast cancer cells MDA-MB-231. Have the authors tested indole derivatives on other TNBC cell lines such as HCC1806, BT20, BT549, SUM159PT, Hs578T or MDA-MB-468 to confirm observed effects? Please refer to the following paper for a non-exhaustive list of TNBC cell lines: https://www.ncbi.nlm.nih.gov/pmc/articles/PMC3532890/

  • Have the authors tested the effects of the four indole derivatives on other cancer cell lines to see if these compounds can be used in other types of cancer than TNBC?

  • Have the authors tested the effects of the four indole derivatives in association with chemotherapeutic agents in TNBC cells MDA-MB-231 in vitro?

  • In figure 3, I am wondering why authors present 2 controls. From the figure 3, I guess that one of the 2 controls corresponds to unlabeled cells (without DCFH-DA). In this case, I don't understand why the intensity of fluorescence is so high (almost 40%) in this control. Please, could the authors explain this result?

  • In the Western-Blot experiment, the signal intensity for Bcl-2 mainly is not uniform along the band of interest. On the other hand, the beta-actin that serves as a loading control appears visually more intense in some samples, which may lead to a bias in the quantification of the BAX/Bcl-2 ratio. Please, could the authors perfom the Western Blot experiment again.

  • Could the authors explain a bit more in the discussion part why differences in chemical structures between the four benzo[f]indole-4,9-dione derivatives lead to differences in biological response, particularly in the generation of ROS by MDA-MB-231 or expression of Bax and Bcl-2 proteins?

Minor comments

  • Line 151: In the legend of Figure 2, LACBio1 is missing.

  • In Figure 1b, I assume that the asterisks of significance are missing.

  • Line 326: please replace “hemotherapeutics” with “chemotherapeutics”.

Author Response

Reviewer 2

Triple-negative breast cancer (TNBC) is an aggressive breast cancer subtype that frequently develops resistance to chemotherapy, metastasizes early despite adjuvant treatment and carries a very poor prognosis. As TNBC has fewer treatment options than other types of invasive breast cancer, there is a critical need to discover new therapeutic agents to treat patients with TNBC.

In this study, Guerra et al. evaluate the action of four benzo[f]indole-4,9-dione derivatives against the TNBC cells MDA-MB 231. Authors show that the four benzo[f]indole-4,9-dione derivative compounds (LACBio1 to LACBio4) can induce cell death by apoptosis through an increase in the generation of ROS, thus causing DNA damage and inducing G2/M cell cycle arrest. Additionally, the authors demonstrate that these compounds exert cytotoxic effects on TNBC by inducing the intrinsic apoptosis pathway, activating BAX-Bcl-2 pathway and caspase 9 in vitro.

Globally, I find this study well designed, well performed and correctly interpreted. However, the following points should be addressed to merit publication in Molecules.

Major comments

  • In this study, the authors treated MDA-MC 231cells with the 4 compounds (at 10 or 30µM) for 24h. Please, could the authors explain why they selected such concentrations to treat the cells. Have the authors performed a viability test (such as a MTT assay) to determine the IC50 of the 4 compounds on these cells?
  • First of all, thank you for your considerations. This is a valid question. We have published a previous study (Dias et al., 2020) showing the synthesis of a range of Benzo[f]indole-4,9-diones derivatives and have screened the cytotoxic assessment by MTT assay of these compounds against a range of human and murine tumor cell lines and non-tumor human erythrocytes or leukocytes. This time, we calculated the IC50 of all synthesized compounds. We chose the best-tested compounds, the 4 used in this study, and tested them against a very aggressive tumor cell line that is unresponsive to current treatments, MDA-MB 231, to investigate the molecular mechanisms involved in cell death induction processes. This information is now explicitly stated in the last paragraph of the introduction and discussed in more depth in the discussion of the paper.

 Dias, F.R.F., Guerra, F.S., Lima, F.A., Castro, Y.K.C., Ferreira, V.F., Campos, V.R., Fernandes, P.D., Cunha, A.C., 2020. Synthesis and biological evaluation of Benzo[f]indole-4,9-diones N-linked to carbohydrate chains as new type of antitumor agents. J. Braz. Chem. Soc. 2020, in press. 

  • The authors demonstrate that the 4 indole derivatives induce apoptosis in MDA-MD 231 cells through an increase in ROS thus causing DNA fragmentation. Have the authors considered and tested other mechanisms involved in cell death, other than apoptosis, such as autophagy, necrosis or ferroptosis?

By using FITC-labeled annexin-V to detect apoptosis by associating propidium iodide, it is possible to measure cells with membrane permeability, it is possible to indicate early and late phases of apoptosis and necrosis (Pec et al., 2003). The Annexin V-FITC/PI binding assay detects live cells (Q2-LL; AV-/PI-), early apoptotic cells (Q2-LR; AV+/PI-), late apoptotic cells (Q2-UR; AV+/PI+) and necrotic cells (Q2-UL; AV-/PI+). So that, we showed in figure 1 that quarter (Q2-UL; AV-/PI+) quite of 0%. This result indicates that the treatment with de Benzo[f]indole-4,9-diones derivatives could not cause necrosis. This information is now explicitly stated in the discussion of the paper.

Pec MK, Aguirre A, Moser-Thier K, Fernández JJ, Souto ML, Dorta J, Diáz-González F, Villar J. Induction of apoptosis in estrogen dependent and independent breast cancer cells by the marine terpenoid dehydrothyrsiferol. Biochem Pharmacol. 2003 May 1;65(9):1451-61. doi: 10.1016/s0006-2952(03)00123-0. PMID: 12732357.

  • The authors show that four benzo[f]indole-4,9-dione derivatives inhibit the growth of triple-negative breast cancer cells MDA-MB-231. Have the authors tested indole derivatives on other TNBC cell lines such as HCC1806, BT20, BT549, SUM159PT, Hs578T, or MDA-MB-468 to confirm observed effects? Please refer to the following paper for a non-exhaustive list of TNBC cell lines: https://www.ncbi.nlm.nih.gov/pmc/articles/PMC3532890/
  • This is a valid and important question. We tested the 4 benzo[f]indole-4,9-dione derivatives in the cell lines MCF-7 (human mammary gland/breast epithelial adenocarcinoma); MDA-MB 231 (human mammary gland/breast epithelial adenocarcinoma); A549 (human lung carcinoma); HT-29 (human epithelial colorectal adenocarcinoma); Hep G2 (human liver hepatocellular carcinoma); SH-SY5Y (human bone marrow neuroblastoma); HT-1080 (human connective tissue epithelial fibrosarcoma) and DMS 79 (human lung pleural fluid carcinoma) and normal human blood peripheral leukocytes and erythrocytes (Dias et al., 2020). The scope of our work was to evaluate the effects of the compounds on different tumor and non-tumor lineages and different cell types. The TNBC MDA-MB 231 cell line was chosen for the continuation of the studies presented in this present work because it was, among the studied lines, the one that led to a worse prognosis for the patient and the one that still did not respond to current treatments. We have not yet tested on another TNBC cell line and we are curious what the results would be, however, we are unaware of any studies that provide the answer.

  • Have the authors tested the effects of the four indole derivatives on other cancer cell lines to see if these compounds can be used in other types of cancer than TNBC?
  • This question is also quite pertinent. As we answered in the previous question, tests performed on other cell lines were published in our previous study (Dias et al., 2020). This information is now explicitly stated in the introduction of the paper.

  • Have the authors tested the effects of the four indole derivatives in association with chemotherapeutic agents in TNBC cells MDA-MB-231 in vitro?
  • As these compounds are new, the intention of this work was to present their individual action. As the molecular mechanisms were still unknown, the scope of this work was to elucidate, even if partially, what were the action targets of these compounds. Now, with the suggestion presented in this work, we can already rationalize an association with other chemotherapeutic agents. This is certainly a very interesting question and we will actively seek the answer in our laboratory.
  • In figure 3, I am wondering why authors present 2 controls. From figure 3, I guess that one of the 2 controls corresponds to unlabeled cells (without DCFH-DA). In this case, I don't understand why the intensity of fluorescence is so high (almost 40%) in this control. Please, could the authors explain this result?
  • Thank you for your considerations. It is a very good question. Autofluorescence is a phenomenon that occurs naturally in a great diversity of cells, due to the presence of different biomolecules, such as NADH, NADPH, flavins, porphyrins, among others (Valentine et al., 2013; Croce & Bottirolli, 2014; Shah et al., 2017; Carver et al., 2019). It is possible to observe a certain pattern of autofluorescence for cells and tissues, due to changes in the levels of these biomolecules, especially in tumor cells. In this case we can call it interfering fluorescence or signal/noise. Another hypothesis that we can present is the possible presence of autofluorescence from dead cells (Liang et al., 2007; Dittmar et al., 2012). 

Valentine, R.M.; Ibbotson, S.H.; Wood, K.; Brown, C.T.A.; Moseley, H. Modelling fluorescence in clinical photodynamic therapy. Photochemical & Photobiological Sciences. v. 12, n. 1, p. 203-213, 2013.

Croce AC, Bottiroli G. Autofluorescence spectroscopy and imaging: a tool for biomedical research and diagnosis. Eur J Histochem. 2014 Dec 12;58(4):2461. doi: 10.4081/ejh.2014.2461. PMID: 25578980; PMCID: PMC4289852.

Shah AT, Cannon TM, Higginbotham JN, Coffey RJ, Skala MC. Autofluorescence flow sorting of breast cancer cell metabolism. J Biophotonics. 2017 Aug;10(8):1026-1033. doi: 10.1002/jbio.201600128. Epub 2016 Oct 12. PMID: 27730745; PMCID: PMC5547001.

Carver GE, Locknar SA, Weaver DL, Stein JL, Stein GS. Real-time detection of breast cancer at the cellular level. J Cell Physiol. 2019 May;234(5):5413-5419. doi: 10.1002/jcp.27451. Epub 2018 Oct 26. PMID: 30362286; PMCID: PMC6344234.

Liang J, Wu WL, Liu ZH, Mei YJ, Cai RX, Shen P. Study the oxidative injury of yeast cells by NADH autofluorescence. Spectrochim Acta A Mol Biomol Spectrosc. 2007 Jun;67(2):355-9. doi: 10.1016/j.saa.2006.07.035. Epub 2006 Jul 31. PMID: 16949859.

Dittmar R, Potier E, van Zandvoort M, Ito K. Assessment of cell viability in three-dimensional scaffolds using cellular auto-fluorescence. Tissue Eng Part C Methods. 2012 Mar;18(3):198-204. doi: 10.1089/ten.TEC.2011.0334. Epub 2011 Dec 14. PMID: 21981657; PMCID: PMC3285599.

  • In the Western-Blot experiment, the signal intensity for Bcl-2 mainly is not uniform along with the band of interest. On the other hand, the beta-actin that serves as a loading control appears visually more intense in some samples, which may lead to a bias in the quantification of the BAX/Bcl-2 ratio. Please, could the authors perform the Western Blot experiment again.
  • A new experiment was performed and the quantification of the BAX/Bcl-2 ratio was done again and included in the text.

  • Could the authors explain a bit more in the discussion part why differences in chemical structures between the four benzo[f]indole-4,9-dione derivatives lead to differences in biological response, particularly in the generation of ROS by MDA-MB-231 or expression of Bax and Bcl-2 proteins?

The compounds LACBio1 and LACBio2, which have an amino group at the C-2 position of the quinone nucleus, were more effective at generating reactive oxygen species in MDA-MB 231cells than the other derivatives tested. This result is in agreement with the data reported in the literature, which indicates that the introduction of oxidizable groups, such as amino substituent, into the quinone ring, can exert an influence on its redox properties. This information is now explicitly stated in the discussion of the paper.

Minor comments

  • Line 151: In the legend of Figure 2, LACBio1 is missing.
  • Done
  • In Figure 1b, I assume that the asterisks of significance are missing.
  • Done

  • Line 326: please replace “hemotherapeutics” with “chemotherapeutics”.
  • Done

Reviewer 3 Report

This paper reports the effects of LACBio1, LACBio2, LACBio3 and LACBio4 (10 and 30 uM) in MDA-MB 231 cells. They measured apoptosis, ROS levels within the cells by using a ROS-sensitive fluorometric probe, DCFH-DA, the Bax (pro-apop-totic) and Bcl-2 (anti-apoptotic) proteins and caspases 3 and 9 . They report that these compounds are able to induce the apoptosis via activation of intrinsic caspase pathways by in-creasing intracellular ROS levels. Finally, they report that the four benzo[f]indole-4,9-dione derivatives induced cell cycle arrest at G2/M phase and have the capacity to inhibit MDA-MB 231 cell proliferation.

These findings could be important for the triple negative breast cancer (TNBC) that is very aggressive, with bad clinical prognosis compared to the of other types of breast cancer.

I have a few questions/ comments:

Q1: Is there a cytotoxic activity against other human cancer cell and also for the normal cells? Are these drugs stable in the cell culture media?

Q2: What was the rational for the design of these compounds (benzo[f]indole-4,9-dione derivatives?

Q3:  Is the proapoptotic pathways activation specific for the MDA-MB 231 cells line or in general? Please Clarify

Q4: How can we explain the similar % of apoptosis observed at all concentrations tested, (22-25% for early apoptosis and 17-21% for late apoptosis of MDA-MB 231 cells within 24 h of treatment (Figure 1A, B). Is there a difference at 48 h?

Q 5:  The control group Fig 1 needs to be clarified. What was exactly this control group?

Q5:  What was the rational for the chosen doses of 10 and 30 uM ? IC50 seems to be less than that. Clarifications are needed. 

Q6: We agree with the statement that these compounds could be promising therapeutic agents because of the induction of apoptosis, TNBC cells (highly proliferative and metabolically active TNBC cells). But what is the effect of these compounds on normal cells?

Q7: Does cell morphology change? If yes, is this dose dependent?

Line 57: It was found that the compounds LACBio1, LACBio2 and LACBio3 increased the total ROS level, and the activation of proapoptotic pathways. What about the LACBio4?

Line 69: “without significant cell death”. How this was tested?

Author Response

Reviewer 3

This paper reports the effects of LACBio1, LACBio2, LACBio3 and LACBio4 (10 and 30 uM) in MDA-MB 231 cells. They measured apoptosis, ROS levels within the cells by using a ROS-sensitive fluorometric probe, DCFH-DA, the Bax (pro-apoptotic) and Bcl-2 (anti-apoptotic) proteins and caspases 3 and 9. They report that these compounds are able to induce apoptosis via activation of intrinsic caspase pathways by increasing intracellular ROS levels. Finally, they report that the four benzo[f]indole-4,9-dione derivatives induced cell cycle arrest at G2/M phase and have the capacity to inhibit MDA-MB 231 cell proliferation.

These findings could be important for the triple negative breast cancer (TNBC) that is very aggressive, with bad clinical prognosis compared to the of other types of breast cancer.

I have a few questions/ comments:

Q1: Is there a cytotoxic activity against other human cancer cell and also for the normal cells? Are these drugs stable in the cell culture media?

First of all, thank you for your considerations. This is a valid question. We have published a previous study (Dias et al., 2020) showing the synthesis of a range of Benzo[f]indole-4,9-diones derivatives and have screened the cytotoxic assessment by MTT assay of these compounds against a range of human and murine tumor cell lines and non-tumor human erythrocytes or leukocytes. This time, we calculated the IC50 of all synthesized compounds. We chose the best-tested compounds, the 4 used in this study, and tested them against a very aggressive tumor cell line that is unresponsive to current treatments, MDA-MB 231, to investigate the molecular mechanisms involved in cell death induction processes. This information is now explicitly stated in the last paragraph of the introduction and discussed in more depth in the discussion of the paper.

Q2: What was the rationale for the design of these compounds (benzo[f]indole-4,9-dione derivatives?

This is a valid question. We have published a previous study (Dias et al., 2020) showing the synthesis of a range of Benzo[f]indole-4,9-diones derivatives. It was indicated in the paper in line 51.

Q3:  Is the pro-apoptotic pathways activation specific for the MDA-MB 231 cells line or in general? Please Clarify

In a previous study (Dias et al., 2020) we showed a range of Benzo[f]indole-4,9-diones derivatives and have screened the cytotoxic assessment compounds against a range of human and murine tumor cell lines and non-tumor human erythrocytes. We chose the best-tested compounds, the 4 used in this study, and tested them against a very aggressive tumor cell line that is unresponsive to current treatments, MDA-MB 231, to investigate the molecular mechanisms involved in cell death induction processes. This information is now explicitly stated in the last paragraph of the introduction and discussed in more depth in the discussion of the paper.

Q4: How can we explain the similar % of apoptosis observed at all concentrations tested, (22-25% for early apoptosis and 17-21% for late apoptosis of MDA-MB 231 cells within 24 h of treatment (Figure 1A, B). Is there a difference at 48 h?

Early apoptotic cells are Annexin V-positive and PI-negative (Annexin V-FITC+/PI), whereas late (end-stage) apoptotic cells are Annexin V/PI-double-positive (Annexin V-FITC+/PI+). Positive labeling with propidium iodide can measure cells with membrane permeability. We evaluated the distribution of cells by the quadrants already described and we could observe that this similar percentage of cell distribution between early and late apoptosis maybe because both in the process of early and late apoptosis, annexin V can cross the permeabilized membrane at the same time as the iodide of propidium enters the cytosol. This shows heterogeneous cell populations.

Q 5:  The control group Fig 1 needs to be clarified. What was exactly this control group?

This control group is formed for untreated cells and stained with annexin V-FITC and propidium iodide. This information is now explicitly stated in the line 67-71.

Q5:  What was the rational for the chosen doses of 10 and 30 uM? IC50 seems to be less than that. Clarifications are needed. 

This question is related to the first question. The hypotheses were clarified in the last paragraph of the introduction.

Q6: We agree with the statement that these compounds could be promising therapeutic agents because of the induction of apoptosis, TNBC cells (highly proliferative and metabolically active TNBC cells). But what is the effect of these compounds on normal cells?

In our previous study (Dias et al., 2020) we tested the 4 Benzo[f]indole-4,9-diones derivatives in normal cells and we could not see cytotoxicity. This information is now explicitly stated in the last paragraph of the introduction and discussed in more depth in the discussion of the paper.

Q7: Does cell morphology change? If yes, is this dose dependent?

This is a very good question. In our previous study (Dias et al., 2020), we showed the morphological changes in MDA-MB 231 cells after treatment with the 4 compounds in a final concentration of 30 µM. We showed apoptotic bodies, shrinkage, membrane blebbing, and losing contact with adjacent cells. We test only at the highest concentration. Minor ones have not been tested. This was one of the biggest indications that the cytotoxic effect we are finding with the Benzo[f]indole-4,9-diones derivatives was due to apoptosis. Thus, we decided in the present study to confirm cell death by apoptosis and the molecular mechanisms involved in this process induced by treatment with Benzo[f]indole-4,9-diones derivatives.

Line 57: It was found that the compounds LACBio1, LACBio2 and LACBio3 increased the total ROS level, and the activation of proapoptotic pathways. What about the LACBio4?

We have shown that the derivative LACBio4 increased reactive oxygen species levels in the cells upon treatment with 10 µM. We suggest that the solubility of LACBio4 at the major concentration, 30 µM, in the ROS generation assay, it could interfere with the results, requiring a time longer than 1 hour of treatment, as established in the assay, for this compound to reach its site of action in the cell. This information is now explicitly stated in the discussion of the paper.

Line 69: “without significant cell death”. How this was tested?

The paragraph has been clarified and changed at line 73.

Round 2

Reviewer 1 Report

I am sorry that I still feel that the significance and the novel of this research was away from the current published level.

Thus, I do not recommend the publication of this paper in Molecules.